# Biotechnological Strategies Adopted for Sugarcane Disease Management in Tucumán, Argentina

**DOI:** 10.3390/plants12233994

**Published:** 2023-11-28

**Authors:** Josefina Racedo, Aldo Sergio Noguera, Atilio Pedro Castagnaro, María Francisca Perera

**Affiliations:** Institute of Agroindustrial Technology of Northwest of Argentine (ITANOA), Obispo Colombres Agroindustrial Experimental Station (EEAOC)—National Council for Scientific and Technical Research (CONICET), CCT NOA Sur. Av. William Cross 3150, Las Talitas, Tucumán T4101XAC, Argentina; joracedo@gmail.com (J.R.); noguera@eeaoc.org.ar (A.S.N.); atiliocastagnaro@gmail.com (A.P.C.)

**Keywords:** *in vitro* culture, hydrothermotherapy, molecular diagnosis, bionanoparticles, bioproducts, markers associated with disease resistance

## Abstract

Sugarcane diseases can be controlled by an integrated management approach where biotechnological tools can successfully contribute. The Obispo Colombres Agroindustrial Experimental Station (EEAOC) in Tucumán (Argentina’s main sugarcane producer) has successfully implemented multiple strategies that greatly enhance the productivity of sugarcane fields. The local breeding program develops resistant varieties by applying molecular markers to reveal the presence of *Bru1* gene for brown rust resistance throughout the EEAOC germplasm collection. In addition, SNP alleles linked to novel sources of resistance were identified following a selective genotyping strategy. Another strategy is the implementation of a seed cane sanitation project using hydrothermal therapy, an *in vitro* culture technique, molecular diagnosis of diseases, and bionanoparticles. As a result, the incidence of systemic diseases has significantly decreased in the production fields. More recently, the use of biological products has shown to be effective for disease control in EEAOC varieties. In summary, several biotechnological strategies including molecular markers associated with resistant sources, *in vitro* culture of apical meristems, molecular diagnostic techniques, and the use of bioproducts are being successfully used for the sustainable management of sugarcane diseases in Tucumán, Argentina.

## 1. Overview of Sugarcane and the EEAOC in Tucumán, Argentina

The production of sugar from sugarcane, one of the oldest agro-industries, constitutes one of the most important economic and social activities in the Northwest region of Argentina. It is mainly concentrated in the province of Tucumán, where 73% of the country’s cane fields are located, being responsible for more than 65% of the total sugar production [1].

Since its creation in 1909, the Obispo Colombres Agroindustrial Experimental Station (EEAOC) in Tucumán has been in charge of importing, evaluating, developing, and distributing most of the sugarcane varieties planted in the province. Likewise, the first sugarcane breeding program in the country was established at the EEAOC in the 1960s, with its own seed production and selection system [1]. Currently, the main objective of the EEAOC breeding program is to develop TUC varieties with improved cane yield, high sucrose content, early maturity, and resistance to the prevalent diseases in the region.

Throughout history, the varieties released by this institution have provided solutions to serious productivity crises, becoming a key tool to achieve substantial increases in the productivity of the local sugar agro-industry. For example, a severe epidemic caused by the sugarcane mosaic virus (SCMV) broke out in Tucumán between 1914 and 1916, affecting all the sugarcane fields and causing losses of around 80% in sugar production. That crisis was effectively solved by replacing the variety cultivated at that time with others introduced from Java (Indonesia) and previously evaluated by the EEAOC. Thus, in 1919, POJ 36 and POJ 213 varieties dominated the Tucuman cane fields, with excellent production levels [2].

Similarly, in 1940 a new disease caused by the fungus *Sporisorioum scitamineum* (sugarcane smut) was detected in Tucumán. POJ 36, the most widespread variety at that time, was highly susceptible to this disease; however, the EEAOC quickly disseminated more appropriate varieties to deal with this new epiphyte [2].

Historically, the control of sugarcane diseases in all production areas has been a major concern due to their impact on productivity. In Argentina, sugarcane is affected by various diseases, such as brown rust (*Puccinia melanocephala*), orange rust (*P. kuehnii*), smut (*Sporisorium scitamineum*), ratoon stunting (RSD, *Leifsonia xyli* subsp. *Xyli*), leaf scald (LS, *Xanthomonas albilineans*), red stripe (*Acidovorax avenae* subsp. *avenae*), sugarcane mosaic (SMV and sorghum mosaic virus, SrMV), and yellow leaf syndrome (SYLV) [1].

In general, in modern sugarcane production systems, diseases are mostly controlled using an integrated approach. This mainly involves the combination of disease-resistant cultivars, disease-free plant material, good management practices, and strict quarantine measures regarding the exchange of foreign plant materials [3]. To achieve this integrated approach, different biotechnological strategies can contribute effectively.

The biotechnological strategies adopted by the EEAOC to contribute to the sustainable management of sugarcane diseases in the region include the obtention of resistant varieties assisted by molecular markers, the use of *in vitro* culture of apical meristems—where the donor plants and the micropropagated plants are evaluated by molecular diagnostic techniques—and the development and adoption of bioproducts to fulfill the requirement of economic and ecological sustainability (Figure 1).

## 2. Molecular Markers in the Breeding Program

Since its beginnings, conventional genetic improvement of sugarcane has had indisputable success, with its main objective being to increase yields (sugar and biomass), followed by resistance to diseases, given the considerable losses that they can cause in the productive system [4]. The use of resistant varieties is one of the fundamental pillars in integrated management, since it constitutes a key tool for sustainable production. However, the genetic complexity of sugarcane (modern cultivars are interspecific, polyploid, and aneuploid) and its semi-perennial cultivation impose great difficulties in the selection of new varieties. Due to these features, many years of intense phenotypic evaluations are required during successive crop cycles, in order to obtain robust characterizations. 

The EEAOC Breeding Program usually performs biparental hybridizations and, to a lesser extent, polycrossings. The recombination event only occurs once initially, and then 11 years of phenotypic evaluations of the progeny are required to identify superior clones. In order to increase the chances of obtaining outstanding genotypes, the EEAOC breeding program performs more than 400 biparental crosses annually, involving more than 100 parents [1]. During the first year of evaluation, 70,000 clones are subjected to phenotypic selection, prioritizing those of good agronomic type and resistant to diseases.

Although in other crops molecular markers are valuable biotechnological tools that assist conventional breeding to improve the precision and efficiency of selection at each stage, marker-assisted selection (MAS) is difficult to implement in sugarcane breeding schemes. First, a prerequisite to implement MAS consists of the availability of markers linked to resistance genes that explain enough percentage of the phenotypic variation observed for the disease resistance. Until now, there were few molecular markers linked to sugarcane disease resistance that meet this characteristic and that have been implemented in breeding programs [5,6]. Secondly, the characteristics of the breeding scheme, with a huge number of initial clones to be evaluated, make the implementation of MAS highly expensive and difficult to perform. Moreover, if breeders are only relying on a limited number of diagnostic resistance markers, they may be excluding high-yielding clones that carry additional resistance sources beyond the ones identified by the markers. This is particularly important considering that the primary breeding aim is high yield. 

Currently available diagnostic markers revealing sources of resistance to sugarcane diseases are useful for studying the frequency and distribution of resistance sources present in the germplasm, even in the absence of disease conditions in the field. They have been widely applied to identify parents carrying specific resistance sources [7,8,9,10,11]. Additionally, a well-planned production system requires not just one but several different resistance sources coexisting in the cultivated area to extend the useful life of such resistance genes. In this regard, molecular markers can also differentiate the resistance alleles present in the region and determine the importance of incorporating new sources into the breeding program.

Sugarcane brown rust is one of the most prevalent diseases in the world, caused by the biotrophic fungus *P. melanocephala*. The main method of controlling it is the use of cultivars resistant to the disease. The discovery of the *Bru1*, a major gene for brown rust resistance in sugarcane identified in the R570 variety [12], marked a turning point for genetic studies and breeding. For the first time in sugarcane, the observed resistance was attributed to a single major gene with a dominant effect, involving a single copy of the resistance allele. The availability of diagnostic molecular markers made it possible to study the distribution of this source of resistance in different germplasm banks around the world [5,7,8,9]. It was observed that resistance often relies on *Bru1*, emphasizing the crucial need to introduce additional sources of resistance. Additionally, some resistant genotypes that did not show *Bru1* diagnostic markers were observed, suggesting the presence of other resistance sources. In the EEAOC Breeding Program, Racedo et al. [13] evaluated sugarcane genotypes from the germplasm bank. It was determined that 15% of the studied accessions presented a resistant behavior to brown rust, but only 6.5% of the cultivars presented the *Bru1* gene. These results allowed to conclude that (i) the *Bru1* gene is efficient to control the brown rust under local conditions of natural infection, (ii) the *Bru1* gene is useful for the development of future varieties, since none of the varieties present in the production system have this gene, and (iii) the potential presence of at least one source of resistance other than the *Bru1* gene was observed. From these results, the local breeding program had incorporated the 21 accessions carrying *Bru1* in the active progenitors, i.e., more frequently used progenitors.

Additionally, the findings prompted the development of the recent study carried out in the EEAOC that allows to identify a set of 34 SNPs linked to resistance to brown rust in sugarcane different to *Bru1* [14]. The selective genotyping methodology carried out with DArT seq markers combined with phenotyping under conditions of natural infection in the field and artificial infection under controlled conditions allowed to locate new sources of resistance derived from the RA 87-3 variety. Those markers explaining 67% of the phenotypic variation are promising for their implementation in the breeding program in order to determine their prevalence in the local germplasm.

The availability of molecular markers linked to different brown rust resistance sources will allow development of a more efficient strategy of marker assisted selection across the selection stages.

## 3. *In Vitro* Culture of Apical Meristems

Taking into account that sugarcane vegetative propagation favors the spread of diseases and that the limited seed multiplication rate leads to slow spread of new cultivars for growers, the EEAOC has implemented thermotherapy, meristem cultures, and micropropagation techniques, within the framework of the sugarcane Vitroplant Project, since 2001. It produces an annual average of 85,000 sugarcane seedlings of the main commercial varieties and promising clones of the final selection stages of the EEAOC’s sugarcane breeding program [1].

The plant tissue culture technique allows the establishment, manipulation, and development of cells, tissues, or organs under aseptic conditions [15], while micropropagation is one of the most important global agricultural biotechnology techniques routinely used for the rapid generation of high-quality, uniform, and disease-free plant material [16]. Different *in vitro* sugarcane micropropagation protocols are available to produce seed canes of high phytosanitary quality (healthy) and with genetic purity (similar to their mother plants) [17]. In the specific case of the EEAOC´s vitroplant project, all seedlings are produced using optimized protocols for each variety where the whole process can be divided into two major phases: (i) meristem culture and micropropagation which include five stages and (ii) molecular analyses [18].

### 3.1. Meristem Culture and Micropropagation of Plant Material

Stage 1, preparation of starting plant material: The multiplicated genotypes are selected each year considering both the demand from sugarcane growers and the advice of EEAOC technicians to broaden the varietal spectrum and reduce the risk of breaking resistance associated with the cultivation of a few elite varieties.

Donor plants are hydro-heat-treated and grown under perfect health and nutritional conditions in greenhouses with natural light and anti-aphid screens. This donor plant collection was implemented in 2006 and is renewed every three years.

Stage 2, establishing the plant culture: The apical meristem is obtained from the apical tip of donor plants. All expanded and encircling leaves are removed, and the cylinders obtained are washed and disinfected. After that, all plant material surrounding the uppermost part of the tip is removed to obtain the apical meristem which is cut and inserted in an inverted position in a tube with solid plant growth medium. The tubes are incubated in darkness for 7 days at room temperature to diminish phenolic oxidation and later transferred to plant growth chambers with a photoperiod of 16:8 h (light: dark) until the formation of shoots, for about 30 days.

Each implanted meristem constitutes a culture line identified with a code to track all future plants originating from this specific meristem.

Stage 3, multiplication of plant material: The first shoot obtained in the previous stage constitutes the starting material. It is transferred to a fresh growth medium containing higher concentrations of the plant hormone cytokinin which induces the formation of new shoots. Newly formed shoots are subdivided into groups of 3–4 and transferred to fresh shoot-inducing media to produce more shoots. Each cycle takes around 30 days and is repeated a maximum of six times to minimize the occurrence of somaclonal variation, yielding a relatively low multiplication rate which generally produces between 1800 and 2000 plantlets at the end of this stage.

Multiplication, where the number of plantlets exponentially increases, is the most time-consuming part of the whole process. Therefore, to multiply only completely healthy plants, molecular diagnosis is performed to evaluate the presence of causal agents of important economic diseases before the second sub-culture of shoots.

On the other hand, in micropropagation, the use of liquid culture media is considered ideal for large-scale production since it reduces manipulations. However, the presence of residual water in the apoplastic spaces causes a physiological disorder known as hyper-hydricity or vitrification [19]. To overcome this inconvenience, one of the best production methods is temporary immersion bioreactors (TIBs) [20]. In this system, all explants are in contact with the culture medium for a very short period of time with a certain daily frequency. Moreover, it is an easy-to-use culture system that allows semi-automation of micropropagation, reducing production costs and increasing multiplication coefficients. Other advantages of these systems are the renewal of the culture medium without changing the culture vessel and the size of the containers that can be larger than those used for solid or semi-solid media [21]. In the EEAOC´s project, TIBs were first implemented for those genotypes with recalcitrant behavior in the conventional micropropagation system; however, currently, they are implemented for any genotype to increase multiplication rates. For example, for the TUC 03-12 variety the multiplication coefficient obtained in the conventional system was 7.03, whereas in the TIBs, it was 30.3 which means that 7 and 30 plants were obtained from each initial meristem, respectively [20].

In order to control contamination by microorganisms, generally antibiotics are added to the growth medium. However, silver nano-materials that exhibit broad spectrum biocidal activity toward bacteria, fungi, viruses, and algae have emerged as an efficient candidate for use in agricultural applications. The mechanisms behind their activity on bacteria are not yet fully elucidated, but it is suggested that they are related with uptaking of free silver ions followed by the disruption of ATP production and DNA replication, formation of reactive oxygen species, and direct damage to cell membranes [22]. Since the synthesis of nanoparticles using microorganisms was classified as an environmentally friendly process [23], silver bio-nanoparticles produced by fungi are being evaluated with encouraging results during the micropropagation stage for microbial contamination control in the EEAOC´s project. They are added to the sterile medium, and some concentrations tested are as effective as the antibiotic. 

Stage 4, root formation: When shoots reach sufficient development, root formation is induced in a growth medium without plant growth hormones, supplemented with high-sugar and low-salt mineral concentrations. This stage, which takes around 30 days, is fundamental to obtain adequate root formation and necessary for the adaptation of plants to *ex vitro* conditions.

Stage 5, growth acclimatization: Plants are removed from the *in vitro* growth jar/tubes and washed to eliminate solid growth medium from roots. Thereafter, they are separated, classified based on plant size, and treated in a fungicide solution.

The growth acclimatization process is initiated by transferring the plants to a disinfected growth substrate, and it is performed in a specially conditioned greenhouse with high relative humidity and low light intensity during the first two weeks to avoid dehydration. After that, light intensity is gradually increased, and humidity is slowly lowered [24].

During this critical stage that normally takes around 90 days, plants change from an *in vitro* heterotrophic growth manner to a photosynthetic and completely autotrophic growth behavior including regulation of its water balance with the external environment. The *in vitro* growth conditions lead to low photosynthetic activity, no or low regulation of stomata, the formation of large intercellular spaces, and a lack of wax formation, which all have to be reverted during the acclimatization stage [25]. 

### 3.2. Molecular Analyses

Plant material is routinely evaluated using different molecular methods at different stages of the first phase to ensure a product free of systemic pathogens and genetically identical to its donor plant.

To guarantee that meristem donor plants and micropropagated seedlings are disease-free, both types of plants are evaluated by molecular diagnosis, a sensitive, rapid, and reproducible choice developed for each important causal agent of disease [26]. Donor plants are evaluated annually whereas micropropagated seedlings are checked after the first sub-culture of shoots at stage 3.

PCR protocols are routinely applied to detect the causal agents of three bacterial diseases: *L. xyli* susbp. *xyli* (ratoon stunting), *X. albilineans* (leaf scald), and *A. avenae* subsp. *avenae* (red stripe), and RT-PCR protocols are applied to detect the viral causal agents of two diseases: SCMV (Sugarcane Mosaic Virus) and SrMV (Sorghum Mosaic Virus), which cause sugarcane mosaic disease and SCYLV (Sugarcane Yellow Leaf Virus) responsible for yellow leaf disease. Since the successful introduction of molecular diagnostics, the incidence of pathogens in the field propagation stages of seed cane has been markedly reduced [27].

Another aspect that must be considered in the massive multiplication of plants through *in vitro* culture is that the new conditions to which the cells are subjected can induce the appearance of undesirable mutational events. These genetic modifications, first described by Larkin and Scowcroft in 1981 [28] as somaclonal variation, are transmitted to regenerated plants and can affect biochemical and morphological characteristics of simple or quantitative inheritance [17]. For that reason, in order to detect and quantify somaclonal variation in the EEAOC´s project propagation scheme, a molecular methodology based on molecular markers is routinely applied as a complement to the phenotypic evaluation in the field. Initially, Amplified Fragment Length Polymorphisms (AFLP) markers were employed for this purpose; however, they were replaced by Target Region Amplified Polymorphism (TRAP) markers since they involve fewer steps and reagents, being cheaper than the previously used [29]. The comparison of the DNA profiles of the micropropagated plants with the donor plant profile makes it possible to detect genetic variations before any phenotypic manifestation can be evidenced. In cases where genetic changes are detected (less than 95% similarity), all genetically distinct plants are destroyed to avoid releasing inadequate material. This evaluation is performed at the growth acclimatization stage when *in vitro* growth has been completed.

After genetic purity is assured by molecular markers, plants are field planted in Basic, Registered, and Certified Nurseries for conventionally propagating before being distributed among sugarcane growers for commercial production [30].

In order to improve both the efficiency in the different stages of the process and the quality of the final product to satisfy sugarcane growers, providing them with seed cane of guaranteed health and genetic purity, several modifications were incorporated into the vitroplant production scheme since its beginning (Figure 2) [20,31,32,33,34,35].

In summary, regarding productivity, efficiency, and safety, propagated plants from meristems present several advantages since, in the short term, old and/or infected materials are replaced by healthy materials of high yield potential [26]. As a result of the EEAOC´s project, the incidence of systemic diseases of economic importance for the region has significantly decreased in the sugarcane production fields by using *in vitro* culture, the micropropagation technique, and molecular diagnosis. For example, the RSD incidence level, measured as number of infected stalks/total number of stalks in the Registered nurseries during the 2014–2018 period, resulted in a maximum value of 0.43%, whereas LS incidence levels were between 0.09% and 0.33%. Moreover, in Certified Nurseries for the same period, RSD incidence levels only reached a maximum value of 0.29% [27], driving a significant improvement in sugarcane crop sanitary conditions and yield. Currently, 73% of the sugarcane area in Tucumán is planted with high quality seed cane obtained by this project [1].

## 4. Use of Bioproducts

Currently, it is possible to obtain healthy food by reducing or replacing synthetic agrochemicals that threaten the health and sustainability of agricultural systems with bioproducts [36]. These bioproducts are based on compounds and/or extracts of microorganisms, insects or plants, or live microorganisms, which produce a beneficial effect on health, promote tolerance to abiotic stress, promote plant growth, and increase crop yield [37]. They are extremely valuable, especially in organic production systems [38]. 

Howler is a commercial bioproduct developed by our institute, formulated from the supernatant of the liquid culture of a strain of the fungus *Acremonium strictum* [39], whose main active ingredient is the defense-inducing protein AsES [40]. Among its main characteristics, it has a low production cost, effectiveness at low concentrations, a long shelf life, tolerance to high temperatures, harmlessness to non-target organisms and the environment, and easy application. Numerous previous studies have demonstrated that the Howler application is effective for the control of diseases of bacterial and fungal origin in numerous monocot and dicot species of commercial importance, both under controlled conditions and in the field [41,42]. Specifically in sugarcane, results obtained in trials with Howler showed a favorable trend in disease control in commercial varieties challenged with the pathogen *A. avenae* subsp. *avenae* which causes red stripe [41]. These encouraging results reinforce the use of new active ingredients that prioritize environmental and social safety, without neglecting their effectiveness in the management of phytosanitary problems, promoting ecological balance.

## 5. Conclusions

In summary, several biotechnological strategies including the implementation of molecular markers associated with disease resistant sources, the *in vitro* culture of apical meristems, the routine application of molecular diagnostic techniques to donor plants and micropropagated lines, and the development and adoption of biological products are being successfully used for the sustainable management of the main sugarcane diseases in Tucumán, Argentina.

## Figures and Tables

**Figure 1 plants-12-03994-f001:**
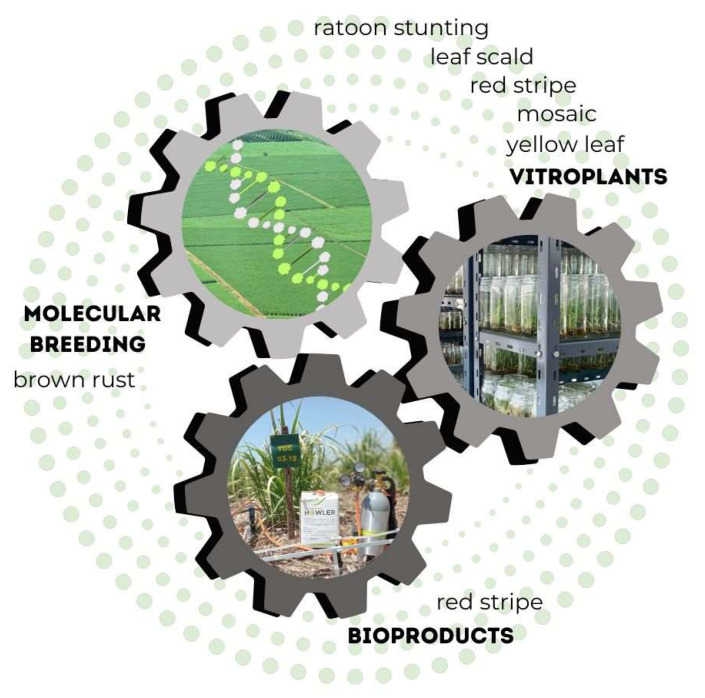
Biotechnological strategies adopted by the EEAOC to contribute to the sustainable management of the main sugarcane diseases in Tucumán, Argentina.

**Figure 2 plants-12-03994-f002:**
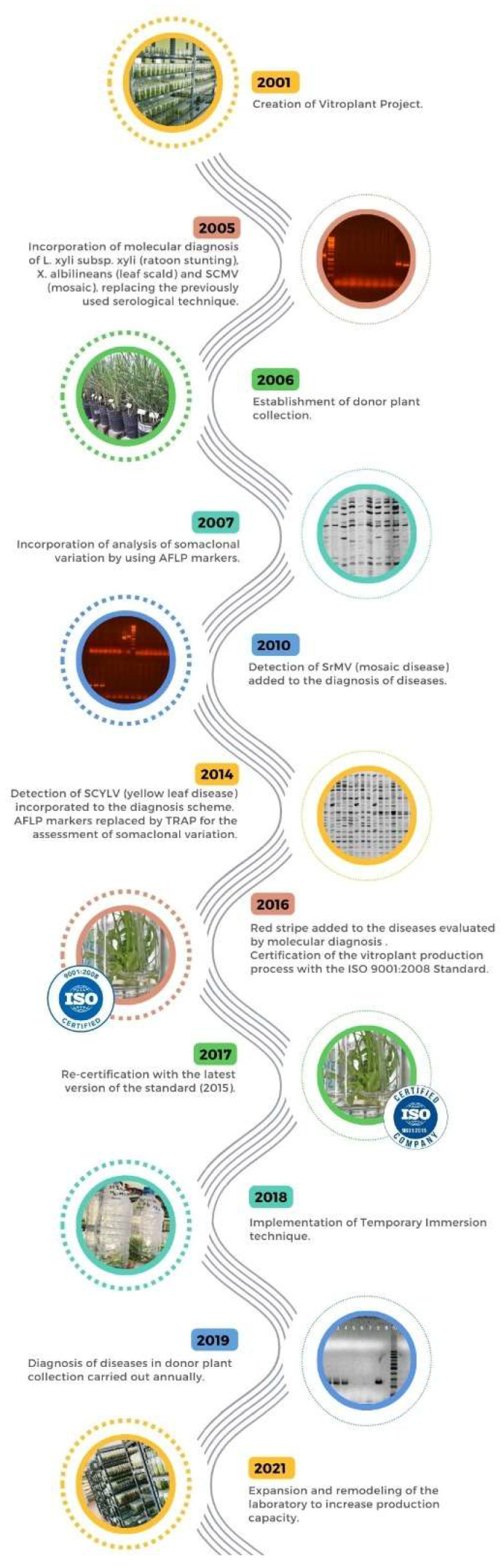
Modifications incorporated in the vitroplant production scheme since its beginning to improve the efficiency in the different stages of the process.

## Data Availability

No new data were created or analyzed in this study. Data sharing is not applicable to this article.

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
