# Peer review of "Biotechnological Strategies Adopted for Sugarcane Disease Management in Tucumán, Argentina"

_plants, 2023, doi:10.3390/plants12233994_

Round 1

Reviewer 1 Report

The manuscript titled "Biotechnological Strategies for Sugarcane Disease Management in Tucumán, Argentina" by Josefina et al appears to be a promising study. However, in its current form, the manuscript lacks a clear focus and contains information that is not relevant to the main topic of the review. To make the review article more informative and improve the flow of information, the authors need to thoroughly revise it.

I recommend that the authors concentrate on either marker-assisted breeding approaches or integrated approaches to managing sugarcane diseases. Currently, the manuscript discusses several different topics, which dilutes its overall focus and makes it less informative. By narrowing the focus, the authors can provide a more in-depth and comprehensive analysis of the chosen approach.

To improve the readability of the manuscript, the authors should ensure that each paragraph has a clear topic sentence and that the information presented is logically structured. The manuscript should also be written in a concise and straightforward manner to ensure that the reader can easily understand the key points.

I am not a native English speaker, Overall language looks ok to me. 

Author Response

Las Talitas, Tucumán, Argentina, 1st June 2023
Dear Ms. Irene Xia
Editor of Plants
I am writing in reference to the manuscript titled: “Biotechnological Strategies for Sugarcane Disease Management in Tucumán, Argentina” by Racedo et al., which is re-submitted for consideration.
Thanks for all the time that you and both reviewers have devoted to improve the manuscript. All suggestions were addressed in the manuscript and below a list of the answers to Reviewer #1 is included.
Reviewer #1:
The manuscript titled "Biotechnological Strategies for Sugarcane Disease Management in Tucumán, Argentina" by Josefina et al appears to be a promising study. However, in its current form, the manuscript lacks a clear focus and contains information that is not relevant to the main topic of the review. To make the review article more informative and improve the flow of information, the authors need to thoroughly revise it.
As it was suggested, the manuscript was thoroughly revised to improve the flow. Some sentences were deleted; even some technical information was removed in order to give the manuscript a clear focus. Besides, the approaches presented were reordered.
I recommend that the authors concentrate on either marker-assisted breeding approaches or integrated approaches to managing sugarcane diseases. Currently, the manuscript discusses several different topics, which dilutes its overall focus and makes it less informative. By narrowing the focus, the authors can provide a more in-depth and comprehensive analysis of the chosen approach.
Marker-assisted breeding approaches were first presented followed by in vitro culture of apical meristems and the use of bioproducts; however, all topics were extensively revised to be more clear, especially the section of bioproducts. The information that could dilute the main focus was removed, especially for the section of in vitro culture and bioproducts, besides a Figure was included to delete information from the text.
To improve the readability of the manuscript, the authors should ensure that each paragraph has a clear topic sentence and that the information presented is logically structured. The manuscript should also be written in a concise and straightforward manner to ensure that the reader can easily understand the key points.
The manuscript was entirely revised according to reviewer´s suggestions. We are confident that the new version fit well the suggestion of reviewer.
Please do not hesitate to contact us for further clarification about any or other issues about the manuscript.
María Francisca Perera

Reviewer 2 Report

Comments to the authors:

This manuscript deals with biotechnological strategies for the sustainable management of sugarcane diseases in Tucumán, Argentina. It is well-written and covered the topic wisely. But, the manuscript is still lacking information related to the status of disease management in sugarcane by biotechnological approaches in omics, and other transgenic approaches. I request the authors to add one or two figures to make it more interesting and informative for the readers. Hence, I recommend a major revision. After making the necessary corrections, the authors can resubmit their manuscript.

Author Response

Las Talitas, Tucumán, Argentina, 1st June 2023
Dear Ms. Irene Xia
Editor of Plants
I am writing in reference to the manuscript titled: “Biotechnological Strategies for Sugarcane Disease Management in Tucumán, Argentina” by Racedo et al., which is re-submitted for consideration.
Thanks for all the time that you and both reviewers have devoted to improve the manuscript. All suggestions were addressed in the manuscript and below a list of the answers to Reviewer #2 is included.
Reviewer #2:
This manuscript deals with biotechnological strategies for the sustainable management of sugarcane diseases in Tucumán, Argentina. It is well-written and covered the topic wisely. But, the manuscript is still lacking information related to the status of disease management in sugarcane by biotechnological approaches in omics, and other transgenic approaches.
The status of disease management in sugarcane by marker-assisted breeding approaches was included, even more, this section was reorder and presented at first place.
Besides the sanitary status in registered and certified nurseries that conventional multiply the micropropagated plants was added in order to show the success of this project.
Regarding transgenic approaches, our research team is working to develop sugarcane varieties resistant to Diatraea saccharalis (sugarcane borer); however, as this topic is focused in disease management, the advances were not included.
I request the authors to add one or two figures to make it more interesting and informative for the readers. Hence, I recommend a major revision. After making the necessary corrections, the authors can resubmit their manuscript.
As reviewer suggested, two figures were added to make the manuscript more interesting for readers.
Please do not hesitate to contact us for further clarification about any or other issues about the manuscript.
María Francisca Perera

Round 2

Reviewer 2 Report

Dear Authors

Thanks for considerable revision of the present manuscript. The present form can be considered for publication.

Good

Author Response

Thanks for your comments.